# Interpretability in Mathematics and XAI

**Alexey Stukachev**
Sobolev Institute of Mathematics
Novosibirsk State University
Novosibirsk, Russia
aistu@math.nsc.ru

Vitaliya Strepetova
Novosibirsk State University
Novosibirsk, Russia
v.strepetova@g.nsu.ru

Liu Chen
Novosibirsk State University
Novosibirsk, Russia
c.lyu3@g.nsu.ru

Zilu Cao
Novosibirsk State University
Novosibirsk, Russia
t.tsao2@g.nsu.ru

Ulyana Zaitseva
Novosibirsk State University
Novosibirsk, Russia
u.penzina@g.nsu.ru

## Abstract

We consider one of the central notions of eXplainable Artificial Intelligence, the notion of interpretability. This notion is actively and fruitfully used in mathematics. We discuss recent results on interpretability of object and processes in mathematical and computational linguistics. These two approaches to understanding and interpreting the meaning of a sentence or a text expressed in natural language provide very typical examples of differences between mathematical (theoretical) and computational (practical) models. Some examples of interpretations of models of one type in models of another type are presented.

## Introduction

We discuss one of the central notions of eXplainable Artificial Intelligence - the notion of interpretability. This notion is actively and and fruitfully used in mathematics: interpreting or representing points as (tuples of) numbers is the central idea of analytic geometry, etc. One of the most general formal definition of interpretability is given in model theory, the branch of mathematical logic studying syntax and semantics of formal languages. Computable model theory studies algorithmic interpretations (representations) of classical mathematical structures on natural numbers. The notion of effective interpretability of abstract structures was studied by many authors. Some of results are collected in survey papers "HF-Computability" (Yu.L.Ershov, V.G.Puzarenko and A.I.Stukachev, 2011)) and "Effective Model Theory: an approach via Sigma-definability" (A.I.Stukachev, 2013). However, objects and processes used in current artificial intelligence are rather different from these studied in mathematical logic (in particular, in model theory, computability theory and proof theory). One of the main features practical objects lack is precision (exactness). We discuss recent results (ours and others) on interpretability of object and processes in mathematical and computational linguistics. These two approaches to understanding and interpreting the meaning of a sentence or a text expressed in natural language provide very typical examples of differences between mathematical (theoretical) and computational (practical) models.

In Section 1, we start with an example of formal mathematical interpetation of the meaning of *negative concord*, tipical for Russian and some other languages (but not for English).

In Section 2, an example of interpretation of formal mathematical model of *Montague semantics* for natural languages in practically oriented model of *DisCoCat*.

In Section 3, we conclude with an example of interpretation (or adaptation) of formal model-theoretical notions of pair and union suitable to be used for *datasets* used in computational linguistics.

# 1 FORMAL SEMANTICS OF NEGATIVE CONCORD: GENERALIZED QUANTIFIERS AND SKOLEM FUNCTIONS

Negative concord refers to the linguistic phenomenon where multiple negative elements in a sentence contribute to a single semantic negation, rather than multiplying negations. The theory of generalized quantifiers is a powerful formal tool for investigating various phenomena in natural language. However, when it comes to the Russian language, a phenomenon such as negative concord may be more effectively studied using a more technical tool, Skolem functions. These functions allow for more precise modelling of the interaction between multiple negative elements, ensuring consistency and avoiding semantic contradictions that may arise when using generalized quantifiers.

## 1.1 GENERALIZED QUANTIFIERS THEORY

The theory of generalized quantifiers, developed in studies Montague (1973), Barwise & Cooper (1981), Keenan & Westerståhl (1997), offers a universal tool for analyzing quantificational expressions. Within this paradigm, quantifiers are interpreted as functions that take predicates and return truth values. For example, the quantifier "someone" can be represented as a function of type $(e \to t) \to t$, where $e$ denotes the type of entities and $t$ denotes the type of truth values. Formally, this is expressed as:

$$[\![\text{someone}]\!] = \lambda P.\exists x.[\text{Human}(x) \wedge P(x)]$$

This method allows for the analysis of statements such as "Some student is buying coffee" through the combination of a quantifier and a predicate. However, in languages with negative concord (e.g., Russian), complications arise: multiple negative elements (such as "no one" and "not") express a single negation. Direct application of generalized quantifiers can lead to semantic contradictions because the formal interpretation ignores the syntactic connection between negative components.

## 1.2 SKOLEM FUNCTIONS IN LINGUISTIC ANALYSIS

Skolem functions provide a formal method for eliminating existential quantifiers. The process of Skolemization involves replacing variables bound by existential quantifiers ($\exists$) with constants or functional symbols, transforming formulas into equivalent forms without existential quantifiers. This approach is crucial for automated theorem proving and the formal analysis of natural language. Consider a formula $\varphi$ of the form:

$$\varphi = \forall y_1 \dots \forall y_n \exists x \psi(x, y_1, \dots, y_n),$$

where $\psi$ is a formula without existential quantifiers. A Skolem function $f$ is introduced to replace the variable $x$, depending on the universally quantified variables $y_1, \dots, y_n$:

$$\varphi_{sk} = \forall y_1 \dots \forall y_n \psi(f(y_1, \dots, y_n), y_1, \dots, y_n)$$

If $n = 0$ (i.e., the existential quantifier does not depend on universal quantifiers), the function $f$ becomes a constant $c$: $\varphi_{sk} = \psi(c)$.

**Example 1**
Original formula: $\forall y \exists x \text{Loves}(y, x)$
After Skolemization: $\forall y \text{Loves}(y, f(y))$,
where $f(y)$ is a Skolem function selecting an object $x$ related by "Loves" to each $y$.

**Example 2**
Original formula: $\exists x \text{Swims}(x)$
After Skolemization: $\text{Swims}(c)$,
where $c$ is a Skolem constant denoting some object for which the predicate "Swims" is true.

Skolem functions enable the formalization of affirmative expressions containing existential quantification semantics. However, they can also be employed in the formalization of Russian language sentences exhibiting negative concord. To achieve this, it is necessary to preprocess such expressions before commencing the direct analysis.

### 1.3 Developing the Algorithm

The formulation of the computational algorithm requires establishing several definitions:

**Definition 1.** A *non-affirmative indefinite pronoun* denotes a non-specific entity marked by the suffix *-nibud'*, which becomes bound within the scope of clausal negation. For instance, *"nikto"* (nobody) exhibits semantic parity with the construction *(neverno, chto + kto-nibud')* (it is not true + somebody) Paducheva (2011).

**Definition 2.** *Aggregated negation* arises when clausal negation merges with an existential quantifier (expressed via *-nibud'* or *hot' odin* phrases in Russian), generating a composite negation marker. This fused *ni*-pronoun obligatorily co-occurs with verbal negation when present Paducheva (2001).

Including these principles, we outline a methodology for transforming Russian negative concord structures into forms amenable to Skolem-based semantic analysis. Crucially, the interaction between sentential negation and indefinite pronouns in such constructions mirrors the behaviour of dedicated negative concord markers. The operational workflow comprises three phases:

1. Detect negative concord patterns by verifying the coexistence of a *ni*-pronoun and a clausal negation particle linked to the predicate.

2. Convert the *ni*-pronoun into its non-specific counterpart (e.g., *kto-nibud'*) while prefixing the proposition with a sentential negation operator (e.g., *"it is false that"*).

3. Eliminate redundant negation markers attached to the verbal complex, preserving only the sentential negation introduced in step 2.

Following these steps, to construct the logical representation of the sentence, it is necessary to introduce a Skolem function to formalize the existence of an object and build a syntactic parse tree, indicating the types of expressions as needed.

## 2 Interpreting Montague Semantics in Distributional Semantics

### 2.1 DisCoCat Model

DisCoCat (Categorical Compositional Distributional) is a mathematical framework developed by B.Coecke, M.Sadrzadeh and S.Clark Coecke et al. (2010) which is used for natural language processing and is based on category theory for a unification of the distributional theory of meaning and a compositional theory.

The syntax of DisCoCat is based on Lambek Pregroups Lambek (1999). They are a recent development and build on his original Lambek (or Syntactic) calculus, where types are used to analyse the syntax of natural languages in a simple equational algebraic setting. The motivation for their use here is that they

share a common structure with vector spaces and tensor products, which are used in the semantics side of this model. Both the category of vector spaces, linear maps, and the tensor product, as well as pregroups, are examples of so-called *compact closed* categories. The meanings of words are vectors in vector spaces, their grammatical roles are types in a Pregroup, and the tensor product of vector spaces paired with the Pregroup composition is used for the composition of (meaning, type) pairs.

## 2.2 MONOIDAL CATEGORIES AND THEIR GRAPHICAL REPRESENTATION

The passage from {true, false}-valuations (as in Montague semantics) to quantitative meaning spaces requires a mathematical structure that can store this additional information, but which at the same time retains the compositional structure. Concrete monoidal categories do exactly that: the axiomatic structure, in particular the monoidal tensor, captures compositionality; concrete objects and corresponding morphisms enable the encoding of the particular model of meaning one uses, here vector spaces.

At first we should briefly recall the basic notions of monoidal category.
A (strict) monoidal category $C$ requires the following data and axioms:

- a family $|C|$ of objects:
    - for each ordered pair of objects $(A,B)$ a corresponding set $C(A,B)$ of morphisms
    - for each ordered triple of objects $(A,B,C)$, each $f : A \to B$ and $g : B \to C$ there is a sequential composite $g \circ f : A \to C$, for which:

    $$(h \circ g) \circ f = h \circ (g \circ f)$$

    - for each object $A$ there is an *identity morphism* $1_A : A \to A$; for $f : A \to B$ we require:

    $$f \circ 1_A = f$$

    $$1_B \circ f = f$$

- for each ordered pair of objects $(A,B)$ a composite object $A \otimes B$, that satisfies:

$$(A \otimes B) \otimes C = A \otimes (B \otimes C)$$

- there is a unit object $I$ which satisfies:

$$I \otimes A = A = A \otimes I$$

- for each ordered pair of morphisms $(f : A \to C, g : B \to D)$ a parallel composite $f \otimes g : A \otimes B \to C \otimes D$; *bifunctoriality* is also required:

$$(g_1 \otimes g_2) \circ (f_1 \otimes f_2) = (g_1 \circ f_1) \otimes (g_2 \circ f_2)$$

What is particularly interesting about monoidal categories is that they admit a graphical calculus in the following sense Selinger (2010): *An equational statement between morphisms in a monoidal category is provable from the axioms of monoidal categories if and only if it is derivable in the graphical language.*

Therefore, next we describe the **graphical language for monoidal categories**.
In the graphical calculus for monoidal categories morphisms are depicted by boxes, with incoming and outgoing wires labelled by the corresponding types, with sequential composition depicted by connecting

matching outputs and inputs, and with parallel composition depicted by locating boxes side by side. For example, the morphisms:

$$1_A \qquad f \qquad g \circ f \qquad 1_A \otimes 1_B \qquad f \otimes 1_C \qquad f \otimes g \qquad (f \otimes g) \circ h$$

are depicted as:

And operations with the unit object I, for example:

$$\psi : I \to A \qquad \pi : A \to I \qquad \pi \circ \psi : I \to I$$

are depicted as:

## 2.3 Higher-order DisCoCat

A new Higher-Order DisCoCat model was introduced by A.Toumi and G.de Felice Toumi & de Felice (2023), where the meaning of a word is not a diagram, but a diagram-valued higher-order function. That allows to give a purely diagrammatic treatment of higher-order and non-linear processes in natural language semantics: adverbs, prepositions, negation and quantifiers. This approach connects the new DisCoCat method with the traditional one — Montague semantics, since it can be seen as a variant of lambda calculus where the primitives act on string diagrams rather than logical formulae.

To better show the logic of HO-DisCoCat model we first should give the definitions:

**Definition 3.** *Monoidal functor* — a functor between monoidal categories that preserves the monoidal structure: a homomorphism of monoidal categories.

**Definition 4.** *Closed functor* — a homomorphism between closed categories.

Montague semantics can be defined as a monoidal functor $F : G \to \Lambda L$, where $G$ is a free closed monoidal category with objects generated by the grammatical types $\{s, n, p\}$ - *sentence, common noun, noun phrase* - together with the words in some finite vocabulary $\omega \in V$, and $\Lambda L$ is a cartesian closed category with objects generated by $L_0 = \{\tau, \phi\}$ - *terms* and *formulae*.

HO-DisCoCat model is defined as a closed monoidal functor
$F : G \to \Lambda D$, where $G$ is a categorial grammar, and $\Lambda D$ is a cartesian closed category generated by $D$ - the language of string diagrams.

We fix a monoidal signature $\Sigma = (\Sigma_0, \Sigma_1, dom, cod)$. Sets $\Sigma_0$ and $\Sigma_1$ are needed for generating *objects* and *boxes*. Whereas $dom, cod : \Sigma_1 \to T$ are functions from boxes to lists of objects $T = \Sigma_0^* = \coprod_{k \in \mathbb{N}} \Sigma_0^k$ also called *types*.

Three main directions to compose diagrams:

1. *left-to-right* using the composition $\circ_{xyz} : (x, y) \times (y, z) \to (x, z)$ which connects the output of one diagram to the input of another

2. *top-to-bottom* using the tensor $\otimes_{xyzw} : (x, y) \times (z, w) \to (xz, yw)$ which concatenates two diagrams side by side

3. *inside-out* using the evaluation $(x, y) \times [(x, y) \to (z, w)] \to (z, w)$ which substitutes one diagram for the free variable in another

The classical DisCoCat model utilises only the first two directions, limiting its ability to express more complex semantic phenomena. In HO-DisCoCat, the third direction—the ability for diagrams to contain other diagrams as embedded structures—enables the modeling of higher-order functions, such as adverbs and other modifiers that influence sentence structure. Next, this additional flexibility will be made more explicit by incorporating it directly into the diagram signature.

The authors Toumi & de Felice (2023) propose to represent second-order processes, such as adverbs, as boxes with k holes that can contain diagrams of a certain shape. We should define *monoidal signature with holes*:

$$\Sigma = (\Sigma_0, \Sigma_1, dom, cod, holes)$$

as a monoidal signature together with a function $holes : \Sigma_1 \to (T \times T)^*$ which assigns to each box $(f : x \to y) \in \Sigma$ a list of pairs of types:

$$holes(f) = ((z_1, \omega_1), \ldots, (z_k, \omega_k)) \in (T \times T)^*$$

for the domain and codomain of the diagrams that can fit in its $k \in \mathbb{N}$ holes.

A practical demonstration of this concept can be seen in the following case: in the sentence "Ideas sleep furiously" the semantics of adverbs like "furiously" can now de defined as boxes with one hole for a verb, prepositions like "with" as boxes with two holes for nouns, etc.

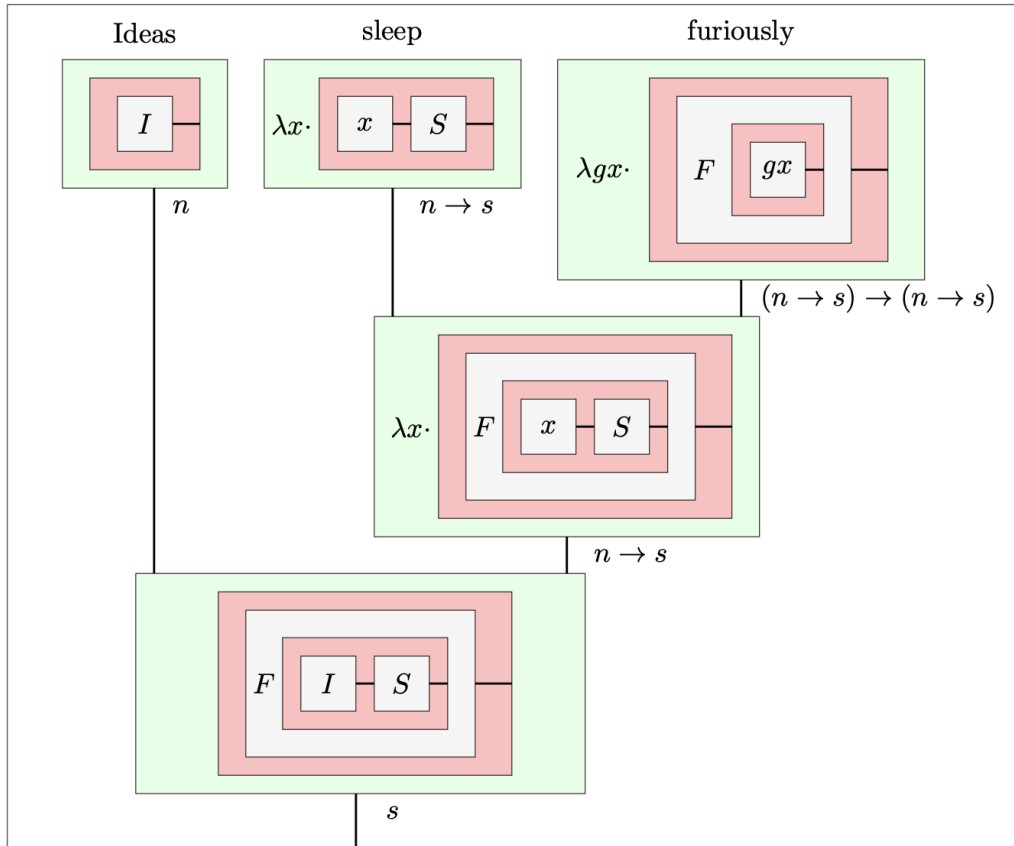

Figure 1: Graphical representation of the sentence "Ideas sleep furiously"

## 2.4 INTERPRETABILITY

As it was already shown in the previous section, combining formal and distributional semantics can lead to better methods of modelling complex linguistic structures. Recently introduced DisCoCirc model Liu et al. (2023) aims to bridge the divide between linguistic theory and modern NLP practice. It represents natural language text as a 'circuit' that captures the core semantic information of the text. These circuits can then be interpreted as modular machine learning models. The semantic and syntactic information taken from the Combinatory Categorial Grammar (CCG) parses of the input text is used in several steps to convert the text into a simply-typed $\lambda$-calculus term, and then into a circuit diagram. Specifically, a step-by-step conversion of $\lambda$-terms into diagrams, including the interpretation of construction rules, is provided in the article Liu et al. (2023).

In the recent article on explainable artificial intelligence (XAI) Tull et al. (2024) it was shown that AI models can be analysed through category theory using a compositional viewpoint, where models are represented as formal string diagrams capturing both their abstract structure and concrete implementation. This comprehensive approach can be applied to a wide range of AI architectures, presenting them as compositional models, ranging from linear and rule-based models, to neural networks, transformers, VAEs, and causal and DisCoCirc models.

By framing interpretability in terms of compositional structure, the approach unifies explainable AI (XAI) concepts and introduces compositionally-interpretable (CI) models, which in addition to linear and rule-based models include, for instance, causal models, conceptual space models, and DisCoCirc models. CI models enhance explainability through diagrammatic methods and rewrite explanations using graphical equations.

## 3 STRUCTURES AND REDUCIBILITIES FOR COMPUTATIONAL LINGUISTICS

As the importance of machine learning continues to increase in various fields, the quality and consistency of datasets have become key issues, especially when computing power is limited. Labelling datasets has been proven to be an effective means to improve training efficiency and accuracy, providing support for improving the accuracy and reliability of data analysis.

The labelling of datasets varies from task to task. This paper specifically studies a structured representation of datasets and their merging methods.

We define the basic components of datasets, including vocabulary, part-of-speech tags, relation tags, and their labelling functions. By introducing contradictory sets, reflexive sets, sequential sets, and general sets, we ensure the integrity and consistency of datasets.

In the process of merging datasets, we describe in detail the construction methods of unions and intersections, and define new part-of-speech and relation labelling functions to process elements from different datasets. In particular, our research provides a theoretical basis for the effective merging of datasets.

### 3.1 INTRODUCTION

For understanding and processing linguistic information, word embedding models and Montague semantics provide different semantic representations: the former captures word relations through vector space and simplifies information using dimensionality reduction techniques, while the latter simplifies semantics through logical structure and reasoning. Reducibilities are typically used to describe the transformation of one problem into another in such a way that known solutions can be utilized to solve the new problem. Through reducibility, we can transform problems in vector spaces into problems in logical reasoning, and vice versa. This article explores how to unify these two semantic representations and provides mathematical methods for their mutual transformation.

A dataset is an organized collection of (linguistic) data. In supervised learning, the labelled target outputs of data instances are key to training models. The labelling content depends on the specific task type. In the field of natural language processing, some datasets label the parts of speech of words Mitchell P. Marcus (1993), and some datasets label the semantic relations between words WordNet official website. Complex models may have multiple labels simultaneously spaCy team. Assuming a dataset has parts of speech and semantic relations as labels, here is the mathematical definition of this dataset:

Let $\mathscr{D}$ be a dataset, then:

$$\mathscr{D} = (V, P, R, L_p, L_r, S)$$

where:

- $V$ is a vocabulary, $V = \{w_1, w_2, \ldots, w_n\}$, $w_i$ represents a word.
- $P$ is a set of part-of-speech tags, $P = \{p_1, p_2, \ldots, p_m\}$, $p_i$ represents a part of speech.
- $R$ is a set of relation labels, $R = \{r_1, r_2, \ldots, r_k\}$, $r_i$ represents a binary relation.
- $L_p : V \to P$ is a part-of-speech labelling function.

- $L_r : V \times V \to \mathscr{P}(R)$ is a relation annotation function.
- $S$ is a set of sentences, $S = \{s_1, s_2, \ldots, s_t\}$.

For any sentence $s \in S$:

$s = (W, E)$

where

- $W = (w_1, w_2, \ldots, w_n)$ is a sequence of words, $w_i \in V$.
- $E = \{(i, j, r) \mid i, j \in [1, n], r \in \mathscr{P}(R)\}$ is a set of relation edges.

and the following two constraints need to be met:

1. Labelling constraints for part of speech: for all words $w \in V : L_p(w) \in \mathscr{P}(P)$
   That is, words may not have part-of-speech labels.
2. Labelling constraints for relations: for all pairs $(w_1, w_2) \in V \times V : L_r(w_1, w_2) \in \mathscr{P}(R)$
   That is, there may be no connection between these two words.

With the set of relation labels, we define four more sets, namely, the contradictory set $A$, the reflexive set $B$, the sequential set $C$ and the general set $G$. This is to ensure the correctness of the dataset. These four sets need to be manually generated based on the relations in the relation label set. We then need to check the dataset against the generated set and make corrections if any inconsistencies are found. The relations contained in the following sets are all from the relation label set.

1. Contradictory set $A$: contains all mutually exclusive pairs of relation labels.
   Here are a few examples:
   - (parent, child): $x$ cannot be both the parent of $y$ and the child of $y$.
   - (greater than, less than): $x$ cannot be both greater than and less than $y$.
   - (true, false): $x$ cannot be both true and false with respect to $y$.
   - (exists, does not exist): $x$ cannot both exist in $y$ and not exist in $y$.

   That is, if $\{r_1, r_2\} \in A$, then if $r_1 \in Lr(w_1, w_2)$, then $r_2 \notin L_r(w_1, w_2)$; or if $r_2 \in L_r(w_1, w_2)$, then $r_1 \notin L_r(w_1, w_2)$.

   For set A, we calculate the number of occurrences of each contradictory relation separately and use it as the deletion level of this relation.

   For example, for set of relation labels $R$, we generate a contradiction set $A$ :

   $$A = \{\{r_1, r_2\}, \{r_1, r_3\}, \{r_1, r_5\}, \{r_2, r_4\}, \{r_3, r_5\}\}$$

   The deletion level of $r_1$ is 3, $r_2$ is 2, $r_3$ is 2, $r_4$ is 1, and $r_5$ is 1 .

2. Reflexive set $B$: contains relation labels that exhibit reflexive properties.
   Here are a few examples:
   - proper subsets: If $x$ is a proper subset of $y$, then $y$ cannot be a proper subset of $x$.
   - Pivot: If $x$ is a pivot of $y$, then $y$ cannot be a pivot of $x$.
   - Greater than: If $a$ is greater than $b$, then $b$ cannot be greater than $a$.
     In addition to independent one-way relations such as proper subsets and pivots, there are also some pairs of one-way relations from the contradictory set.

   That is, if $r \in B$, then if $r \in L_r(w_1, w_2)$, then $r \notin L_r(w_2, w_1)$.

---

**Algorithm 1** Modification algorithm 1

---

1: **for** all pairs $\{r_x, r_y\} \in A$ **do**
2:    **if** find that $r_x \in L_r(w_i, w_j)$ and $r_y \in L_r(w_i, w_j)$, where $r_x, r_y \in R; w_i, w_j \in V$ **then**
3:       Compare the deletion index of $r_x$ and $r_y$
4:       **if** deletion index of $r_x <$ deletion index of $r_y$ **then**
5:          Delete $r_y$ from $L_r(w_i, w_j)$
6:       **else if** deletion index of $r_x >$ deletion index of $r_y$ **then**
7:          Delete $r_x$ from $L_r(w_i, w_j)$
8:       **else**
9:          Randomly delete $r_x$ or $r_y$ from $L_r(w_i, w_j)$
10:       **end if**
11:    **end if**
12: **end for**

---

**Algorithm 2** Modification Algorithm 2

---

1: **for** all relations $r_x \in B$ **do**
2:    **if** find that $r_x \in L_r(w_i, w_j)$ and $r_x \in L_r(w_j, w_i)$, where $r_x \in R; w_i, w_j \in V$ **then**
3:       Randomly delete $r_x \in L_r(w_i, w_j)$ or $r_x$ from $L_r(w_j, w_i)$
4:    **end if**
5: **end for**

---

3. Sequential set $C$: Some relation labels are assigned a priority value, where lower values indicate higher priority. For example:

- The priority of 'proper subset' is 1.

- The priority of 'subset' is 2.

If A is a proper subset of B, then A is also a subset of B.

We can use a pair (proper subset, subset) to represent such an order relation. A set consisting of several such n-ary ordered pairs is called a sequential set.

That is, if $(r_1, ..., r_z) \in C$, then if $r_i \in L_r(w_1, w_2)$, then for all $j \in (i, z]$, $r_j \in L_r(w_1, w_2)$.

---

**Algorithm 3** Modification Algorithm 3

---

1: **for** all sequences $(r_1, \ldots, r_z) \in C$ **do**
2:    **for** each relation $r_i$ in $(r_1, \ldots, r_z)$ **do**
3:       **for** all $r_i \in L_r(w_i, w_j)$ **do**
4:          Add all relations in the sequence which after $r_i$ but not in $L_r(w_i, w_j)$ to $L_r(w_i, w_j)$.
5:       **end for**
6:    **end for**
7: **end for**

---

4. General set $G$: The set consisting of all elements in the relation label set that do not appear in the above three sets.

## 3.2 OPERATION

Integrating datasets from different sources can provide a more comprehensive view of the data and enhance the model's capabilities. During the merging process, conflicting data needs to be cleaned Hong Hai Do

(2000). The article Rutian Liu (2020) introduces a method for dataset merging that utilizes hierarchies to assist in the integration. Based on these concepts, a method for dataset merging is proposed.

For the processed datasets $D_1$ and $D_2$, we can use the definition to describe their union and intersection without any contradiction:

$$\mathscr{D}_1 = (V_1, P_1, R_1, L_{p1}, L_{r1}, S_1)$$
$$\mathscr{D}_2 = (V_2, P_2, R_2, L_{p2}, L_{r2}, S_2)$$

## UNION

To get the union of datasets $D_1$ and $D_2$, we first define the following set $\mathscr{D}$:

$$\mathscr{D} = (V_1 \cup V_2, P_1 \cup P_2, R_1 \cup R_2, L_p, L_r, S_1 \cup S_2)$$

in:

- $L_p$ is a new part-of-speech tagging function, defined as:

$$L_p(w) = \begin{cases} L_{p1}(w) & \text{if } w \in V_1 \text{ and } w \notin V_2 \\ L_{p2}(w) & \text{if } w \in V_2 \text{ and } w \notin V_1 \\ L_{p1}(w) \cup L_{p2}(w) & \text{if } w \in V_1 \text{ and } w \in V_2 \end{cases}$$

- $L_r$ is a new relation annotation function defined as:

$$L_r(w_x, w_y) = \begin{cases} L_{r1}(w_x, w_y) & \text{if } w_x, w_y \in V_1 \text{ and } w_x, w_y \notin V_2 \\ L_{r2}(w_x, w_y) & \text{if } w_x, w_y \in V_2 \text{ and } w_x, w_y \notin V_1 \\ L_{r1}(w_x, w_y) \cup L_{r2}(w_x, w_y) & \text{if } w_x, w_y \in V_1 \text{ and } w_x, w_y \in V_2 \\ \emptyset & \text{otherwise} \end{cases}$$

For the obtained dataset $\mathscr{D}$, we still manually generate the corresponding contradiction set $A$, reflexive set $B$, sequential set $C$ and general set $G$ based on the newly obtained relation label set $R$. The reflexive set $B$ can directly take the corresponding union, but the remaining relations need to be manually identified.Then, according to the generated set, the dataset $\mathscr{D}$ is subjected to the modification algorithm mentioned above to obtain the dataset $\mathscr{D}'$, which is the union of $\mathscr{D}_1$ and $\mathscr{D}_2$.

Similarly, datasets can also take intersections to extract information that is common to both.

## INTERSECTION

The intersection of datasets $\mathscr{D}_1$ and $\mathscr{D}_2$, $\mathscr{D}_1 \cap \mathscr{D}_2$, can be defined as:

$$\mathscr{D}_1 \cap \mathscr{D}_2 = (V_1 \cap V_2, P_1 \cap P_2, R, L_p, L_r, S_1 \cap S_2)$$

in:

- $R$ is a new set of relation labels defined as:

$$R = (R_1 \cap R_2) \cup R'$$

where $R'$ is the set of the first $i$ elements from the same dataset ($\mathscr{D}_1$ or $\mathscr{D}_2$) in all n-ary ordered pairs in sequential set $C$ ($\mathscr{D}_1 \cup \mathscr{D}_2$) that satisfy the following conditions:The n-ary ordered pair must contain elements from both $R_1$ and $R_2$ .

- $L_p$ is a new part-of-speech tagging function defined as:

$$L_p(w) = L_{p1}(w) \cap L_{p2}(w); w \in V_1 \cap V_2$$

- $L_r$ is a new relation annotation function defined as:

$$L_r(w_x, w_y) = \begin{cases} L_{r1}(w_x, w_y) \cap L_{r2}(w_x, w_y) & \text{if } w_x, w_y \in V_1 \cap V_2 \\ L_{r1}(w_x, w_y) \cap L_{r2}(w_x, w_y) \cup r & \text{if } w_x, w_y \in V_1 \cap V_2 \text{ and } r \in R' \\ & \quad \text{and } (r \in L_{r1}(w_x, w_y) \text{ or } r \in L_{r2}(w_x, w_y)) \\ \emptyset & \text{otherwise} \end{cases}$$

## 4  CONCLUSION

In all examples from mathematical and computational linguistics considered in this paper, interpretations can be viewed either as explanations and formalizations (of practical issues), or as adaptations and applications (of theoretical issues). We believe the study of these two types of interpretations is very important for current and future researches in eXplainable Artificial Intelligence. At present, we believe that the main problem is the lack of adequate mathematical and computational models for such studies.

## 5  APPENDIX: Σ-DEFINABILITY

We recall some formal mathematical notions of effective interpretability (Σ-definability) of abstract structures, for which the above notions for datasets can be viewed as *modifications* and *adaptations* for practical tasks from computational linguistics. For more information on this topic see, for example, the survey paper "Effective Model Theory: an approach via Sigma-definability" (A.I.Stukachev, 2013). In our forthcoming paper, these notions are used to formalise and prove (or explain) results on algorithmic complexity for datasets. In particular, we analyse the complexity of the union of datasets by means of semilattices of Σ-degrees of structures. To do this, we need all the definitions below.

- Hereditarily finite sets
  For a set $M$, consider the set $HF(M)$ of hereditarily finite sets over $M$, defined as follows:

$$HF(M) = \bigcup_{n \in \omega} HF_n(M), \text{ where}$$
$$HF_0(M) = \{\emptyset\} \cup M,$$
$$HF_{n+1}(M) = HF_n(M) \cup \{a \mid a \text{ is a finite subset of } HF_n(M)\}.$$

- Hereditarily finite extension
  For a structure $\mathfrak{M} = (M, \sigma^{\mathfrak{M}})$ (finite or computable) of signature $\sigma$, the hereditarily finite extension

$$\mathbb{HF}(\mathfrak{M}) = (HF(\mathfrak{M}); \sigma^{\mathfrak{M}}, U, \in, \emptyset)$$

  is a structure of signature $\sigma'$ (with $\mathbb{HF}(\mathfrak{M}) \models U(a) \leftrightarrow a \in M$). Let $\sigma' = \sigma \cup \{U^1, \in^2, \emptyset\}$, where $\sigma$ is a finite signature.
  Hereditary finite extensions are the simplest examples of admissible sets.

- $\Delta_0$-formulas and $\Sigma$-formulas

  1. The class of $\Delta_0$-formulas of signature $\sigma'$ is the smallest class of formulas containing all atomic formulas of signature $\sigma'$ and closed under $\wedge$, $\vee$, $\neg$, $\exists x \in y$, and $\forall x \in y$.

  2. The class of $\Sigma$-formulas of signature $\sigma'$ is the smallest class of formulas containing all $\Delta_0$-formulas of signature $\sigma'$ and closed under $\wedge$, $\vee$, $\exists x \in y$, $\forall x \in y$, and $\exists x$.

- $\Sigma$-Definability

  Let $\mathfrak{M}$ be an algebraic structure of a computable predicate signature $\langle P_0^{n_0}, \ldots, P_k^{n_k}, \ldots \rangle$, and let $\mathbb{A}$ be an admissible set. The structure $\mathfrak{M}$ is called $\Sigma$-*defined in* $\mathbb{A}$, or $\mathbb{A}$-*constructible*, if there exists a computable sequence of $\Sigma$-formulas

  $$\Phi(x_0, y), \Psi(x_0, x_1, y), \Psi^*(x_0, x_1, y), \Phi_0(x_0, \ldots, x_{n_0-1}, y),$$

  $$\Phi_0^*(x_0, \ldots, x_{n_0-1}, y), \ldots, \Phi_k(x_0, \ldots, x_{n_k-1}, y), \Phi_k^*(x_0, \ldots, x_{n_k-1}, y), \ldots$$

  of signature $\sigma_{\mathbb{A}}$ and a parameter $a \in A$ such that for $M_0 \leftrightharpoons \Phi^{\mathbb{A}}(x_0, a)$ and $\eta \leftrightharpoons \Psi^{\mathbb{A}}(x_0, x_1, a) \cap M_0^2$ the following holds: $M_0 \neq \varnothing$, $\eta$ is a congruence relation on the structure

  $$\mathfrak{M}_0 \leftrightharpoons \langle M_0; P_0^{\mathfrak{M}_0}, \ldots, P_k^{\mathfrak{M}_0}, \ldots \rangle,$$

  where, for all $k \in \omega$, $P_k^{\mathfrak{M}_0} \leftrightharpoons \Phi_k^{\mathbb{A}}(x_0, \ldots, x_{n_k-1}) \cap M_0^{n_k}$, $\Psi^{*\mathbb{A}}(x_0, x_1, a) \cap M_0^2 = M_0^2 \setminus \Psi^{\mathbb{A}}(x_0, x_1, a)$, $\Phi_k^{*\mathbb{A}}(x_0, \ldots, x_{n_k-1}, a) \cap M_0^{n_k} = M_0^{n_k} \setminus \Phi_k^{\mathbb{A}}(x_0, \ldots, x_{n_k-1})$, and the structure $\mathfrak{M}$ is isomorphic to the quotient structure $\mathfrak{M}_0 / \eta$.

- $\Sigma$-Reducibility

  For structures $\mathfrak{A}$ and $\mathfrak{B}$, we denote by $\mathfrak{A} \leqslant_\Sigma \mathfrak{B}$ the fact that the structure $\mathfrak{A}$ is $\Sigma$-defined in $\mathbb{HF}(\mathfrak{B})$.

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
