# OpenReview forum: "Interpretability in Mathematics and XAI"
_mathai.club/MathAI/2025/Conference — MathAI 2025 Oral_

### Official Review · Reviewer_TFKt · 2025-02-26
**INTERPRETABILITY IN MATHEMATICS AND XAI**

**Rating:** 4
**Confidence:** 4

**Review:**

The article has too general title. It would be suitable if the article considered all the variants for defining interpretation in mathematics and XAI, rather than special cases.

In the particular cases presented, it is impossible to understand whether we are talking about known results or new results of the authors, since both results are mixed.

It is unclear how section 1 is related to sections 2 and 3 and whether it is needed. For example, section 2 can also be considered as preliminaries for section 3.

It is unclear what goal the authors of the article set and what problem they solved.

There is no conclusion.

There are a small number of typos.

Line 33: One of the main features practical objects lack - пропущены предлоги

Line 41: tipical

Line 43: There is no predicate in this sentence.

Line 46: unionsuitable

Line 107 and below: Words 'nibud’, 'nikto' and so on must be accompanied by their English counterparts.

Line 214: nd ΛL

Line 303: simultaneouslyspaCy team.

Line 339: data set

Line 384: Starting from

---

### Official Review · Reviewer_1DVF · 2025-02-26
**The article is good overall, but needs some improvements**

**Rating:** 6
**Confidence:** 4

**Review:**

This paper is about research in Natural Language Processing (NLP). It looks at ways to improve how easy it is to understand the structure of natural language sentences. It uses different methods of model theory and lambda calculus.The paper is exactly in line with the conference theme.

It is important to note that in the context of artificial intelligence, the notion of interpretability is much broader than what is discussed in the article in the context of interpretability of objects and processes in mathematical and computational linguistics.Therefore, the title of the paper does not fully reflect its content and should be revised to narrow down to the area under consideration: mathematical linguistics.

The paper is structured as follows: an introduction and three paragraphs. Each paragraph discusses an example of logical interpretability for a certain phenomenon of mathematical linguistics.The paper does not have a Conclusion. This is important because we would like to see a generalisation of the considered examples, findings and a brief formulation of the results of the paper.

---

### Official Review · Reviewer_YhY9 · 2025-02-27
**The paper provides a relevant exploration of interpretability in mathematics and XAI, but it would benefit from a more explicit discussion of the relationship between the theoretical and practical approaches to interpretability**

**Rating:** 7
**Confidence:** 4

**Review:**

Review
The study focuses on certain aspects of the problem of interpretability in mathematics and explainable artificial intelligence (XAI). It examines three different approaches to formalizing the concept of interpretability used in mathematics and discusses their application in mathematical and computational linguistics (separately). An attempt is made to connect theoretical (mathematical) and practical (computational) models of interpreting the meaning of text or sentences in natural language.
Relevance of the Topic
The formalization of interpretability concepts in mathematics and XAI is relevant, as the explainability of artificial intelligence models is becoming an important requirement in various applications, including natural language processing (NLP). The increasing complexity of neural network-based machine learning models necessitates the development of methods that can explain their decisions. Additionally, the use of mathematical approaches to interpret linguistic phenomena, such as negative concord, is also an important direction in modern linguistics and NLP. The article consists of three independent parts: 1. The use of the Skolem function to analyze negative concord in the Russian language, which helps avoid semantic contradictions that arise when using generalized quantifiers. 2. A discussion of the DisCoCat model, which combines category theory and distributed semantics, allowing for more effective modeling of complex linguistic structures. 3. An examination of methods for combining and intersecting datasets to improve the quality and consistency of data in NLP. The DisCoCat model discussed in Section 2.3 appears promising, as it provides a framework for modeling complex semantic phenomena. However, the results of its practical application are not analyzed.
Recommendations for Improving the Article
The explanation of the DisCoCat model in Section 2 should be more clearly connected with graphical illustrations, as they seem somewhat detached from the main text. It would be desirable to clarify the obtained results with examples, such as their applicability to currently popular LLMs, indicating the limitations of applicability and the potential to overcome current limitations in the future. Additionally, it would be useful for understanding the content of the work to include (in the introduction or conclusions) the authors' views on the relationship between the described approaches to the problem of formalizing interpretability.

---

### Decision · Program_Chairs · 2025-03-08

**Decision:**

Accept (Oral)

**Comment:**

Your article has been accepted and you can give a talk on the article. All articles will be sorted by rating and within the available conference places one author from each article will be invited. If there are not enough places, then you will either have the opportunity to speak remotely or come at your own expense!